# Impacts of Space Restriction on the Microstructure of Calcium Silicate Hydrate

**DOI:** 10.3390/ma14133645

**Published:** 2021-06-30

**Authors:** Yue Zhou, Zhongping Wang, Zheyu Zhu, Yuting Chen, Linglin Xu, Kai Wu

**Affiliations:** 1School of Materials Science and Engineering, Tongji University, 4800 Cao’an Road, Shanghai 201804, China; 1710816@tongji.edu.cn (Y.Z.); wangzpk@tongji.edu.cn (Z.W.); 17712937215@163.com (Z.Z.); 1930649@tongji.edu.cn (Y.C.); wukai@tongji.edu.cn (K.W.); 2Key Laboratory of Advanced Civil Engineering Materials, Tongji University, Ministry of Education, 4800 Cao’an Road, Shanghai 201804, China

**Keywords:** calcium silicate hydrate (C-S-H), space restriction, cement, hydration

## Abstract

The effect of hydration space on cement hydration is essential. After a few days, space restriction affects the hydration kinetics which dominate the expansion, shrinkage and creep of cement materials. The influence of space restriction on the hydration products of tricalcium silicate was studied in this paper. The microstructure, morphology and composition of calcium silicate hydrate (C-S-H) were explored from the perspective of a specific single micropore. A combination of Raman spectra, Fourier transform infrared spectra, scanning electron microscopy and energy dispersive X-ray spectroscopy were employed. The results show that space restriction affects the structure of the hydration products. The C-S-H formed in the micropores was mainly composed of Q^3^ silicate tetrahedra with a high degree of polymerization. The C-S-H formed under standard conditions with a water to cement ratio of 0.5 mostly existed as Q^2^ units. Space restriction during hydration is conducive to the formation of C-S-H with silica tetrahedra of a high polymerization degree, while the amount of water filling the micropore plays no obvious role on the polymeric structure of C-S-H during hydration.

## 1. Introduction

As the dominating hydration product of Portland cement, calcium silicate hydrate (C-S-H) plays a key role in the performance of cementitious materials. C-S-H grows when the cement is mixed with water [1], and this hydration continues until the water is exhausted [2]. Its structure varies with factors during hydration, especially the water to binder ratio [3,4], temperature [5,6,7,8] and curing age [6,9]. The growth of C-S-H involves the competition between water and hydration space, which are also essential to the microstructure evolution. The importance of water to the structure of C-S-H is prominent, and the role of space on the structure of C-S-H has been paid more and more attention, especially for those studies on the kinetics and mechanisms of cement hydration.

As the main constituent of Portland cement, the hydration of alite (C_3_S) is essential and it can be divided into three periods [2,10]. Initially, C_3_S massively dissolves and amorphous nanoglobular C-S-H mainly forms in this period (Period I) [11]. There is sufficient space at this stage. Then the hydration switches to the main hydration period (Period II). The C-S-H explosively grows and the hydration rate experiences a stage of acceleration to deceleration [2]. The initial space is continuously filled with hydration products. Initially, C-S-H grows rapidly to a certain length on the cement grains as needle morphology [12]. When it has covered most of the grain surface, this stage is related to the deceleration stage [12]. Then the hydration comes to period III, the later stages [13]. The C-S-H around the grain surface forms a shell which controls the inner space and available water outside. The growth of inner C-S-H is accompanied by a lack of space and water. Meanwhile, the outer C-S-H further fill the space at this stage [14]. The space filling hypothesis is suggested to dominate the later hydration mechanism [2]. The C-S-H which forms in the later stage grows in a diminishing, restricted space. In general, those unhydrated phases undergo a state transition from sufficiency to a restriction of space.

The growth of C-S-H and available space are inseparable. Recent research has proposed that the initial space occupied by water is the space where the C-S-H can grow [2]. The ^1^H nuclear magnetic resonance is used to divide the “water space” in the cement paste into three types: interlayer water between C-S-H layers, gel pores and capillary pores [15,16]. The growth of hydrates in the capillary space is related to the expansion of the hardened cement matrix, due to crystallization pressure [10,17]. The crystals grown in the pores and also the pore walls generate stress to prevent crystal growth. Eventually, the crystals exert pressure on the surrounding solid phase [17]. The pore space affects the shrinkage and creep of the cement paste at the late hydration stage [18,19]. Thus space is considered a key factor for the construction of the C-S-H microstructure model. The classical C-S-H model proposed by Jennings describes two types of C-S-H: low-density C-S-H (LD C-S-H) and high-density C-S-H (HD C-S-H) [20,21]. LD C-S-H forms in the early hydration stage when there is enough capillary space to accommodate C-S-H [22]. With the proceeding of hydration, the continuous filling by hydration products leads to more restriction in the space, which drives LD C-S-H to HD C-S-H [22,23]. This finding indicates that the space limitation is firmly related to the diversity of C-S-H. Nanoindentation results show that HD C-S-H exhibits higher stiffness [24]. The difference in performance actually reflects the change in the microstructure of C-S-H.

The microstructure of C-S-H is commonly expressed by the polymerization degree of silicate tetrahedra. The Q^n^ classification is used to represent the Si-environments in silicate tetrahedra with different polymerization degrees, where n stands for the number of oxygen atoms shared between the tetrahedra and its adjacent tetrahedra [25,26]. Studies show that the silicate tetrahedra of C-S-H in cement are mainly dimers (Q^1^ Si sites), with few trimer (Q^2^ Si sites) [27,28,29]. The silicate tetrahedra in tobermorite and jennite, which are used as reference when constructing the C-S-H model, with repeating structure of “dreierkette” and the ratio of Q^2^ and Q^3^ units are much higher than those of Q^1^ units [27,28]. Studies indicate that the polymerization degree of C-S-H increases with curing age [6,30,31], while the available space decreases over time, simultaneously. There are many mysteries between the formation of highly polymerized units and space restriction. 

During the entire hydration, the “water space” is gradually filled by the growing hydrates and undergoes a process from micrometer to nanometer scale. Therefore, the influence of the space restriction on the structure of C-S-H is explored from the perspective of a single micropore in this paper. 

## 2. Materials and Methods

### 2.1. Sample Preparation

The pure M3 tricalcium silicate (C_3_S) was prepared by calcination. Magnesium was used to stabilize C_3_S with polymorph M3. The amount of chemically pure calcium oxide (CaO) and silicon dioxide (SiO_2_) with a molar ratio of 3.0 was accurately weighed, and MgO was added in an amount of 2% of the total mass. The raw materials were uniformly mixed and put into a platinum crucible. Then the crucible was put into the furnace and calcinated at 1400 °C for 4 h. After that, the sample was cooled to room temperature and ground into powder. To obtain the pure M3 polymorph C_3_S, this treatment was repeated 3 times. The phase composition was analyzed by XRD (Rigaku International Corporation, Akishima, Tokyo, Japan) and the pattern is shown in Figure 1.

The experiment was based on a stainless steel sheet customized by Shanghai Guangyan Laser Technology Co. (Room 896, No.235 Changyang Road, Hongkou District, Shanghai, China), Ltd. (the size was 5 mm × 5 mm, and the thickness was 100 μm). A 4 × 5 pore array was made on the sheet by a 50 w optical fiber marking machine, as shown in Figure 2. The diameter of the pores was 10 ± 5 μm, and these pores did not penetrate the sheet.

C_3_S particles were placed in micropores (Shanghai Guangyan Laser Technology Co, Shanghai, China) of the sheet with a tweezer under a microscope. These samples were treated by two methods. On the one hand, the sheet was covered by a drop of deionized water and was placed at 20 °C and 65% relative humidity (RH) for 24 h (hereafter called water condition). On the other hand, the sheet was placed in the curing room at 20 °C and 65% RH for 24 h without further addition of water (hereafter called 65% RH condition). Meanwhile, C_3_S was mixed with deionized water at a water to cement ratio of 0.5 and hydrated for 24 h under the same curing conditions, as the control. The hydration was terminated via the freeze-vacuum drying for 48 h.

### 2.2. Analytical Methods

Raman spectra were collected by the LabRAM HR Evolution Raman Spectrometer (HORIBA Jobin Yvon, Paris, France) from HORIBA Jobin Yvon, France. A solid-state laser with a wavelength of 532 nm was used as the light source and the acquisition time was 10 s. The spectra were obtained with a 100× magnification. A micro area in a single pore of the sheet was selected for testing. Five pores were randomly selected for testing.

Fourier transform infrared spectra (FT-IR) were recorded using the EQUINOX 55 Fourier transform infrared spectrometer (Bruker, Karlsruhe, Baden-Württemberg, Germany) from Bruker and the MIRacle ATR measurement accessory (PIKE Technologies, Madison, WI, USA) from PIKE Technologies, USA. The ATR sample cell was made of ZnSe crystal. The sheet was placed in the sample cell and all pores were tested simultaneously. Each spectrum was acquired with 128 scans and at a resolution of 4 cm^−1^.

The morphology of samples was obtained by the Gemini Sigma 300/VP scanning electron microscope (SEM) (Carl Zeiss, Oberkochen, Baden-Württemberg, Germany) from Carl Zeiss, Germany. The images were acquired employing an acceleration voltage of 30 kV under high vacuum conditions. The elemental analysis was collected by the X-Max^N^ spectrometer (Oxford Instruments, Abingdon, Oxfordshire, UK) of Oxford Instruments (EDS) and combined with Aztec software (Version 3.2, Oxford Instruments, Abingdon, Oxfordshire, UK).

## 3. Results

### 3.1. Raman

Figure 3 shows the Raman spectra of the hydration products in different pores which were treated by the water conditions. Besides those shifts at 316, 469, 1449, 1510 and 2425 cm^−1^ from the sheet, there was a strong and sharp peak at 1084 cm^−1^ in all pores. This peak may be assigned to the symmetrical stretching of Q^3^ silicate tetrahedra, or the symmetric stretching of the C-O group in CaCO_3_ [27,32]. The shift at 280 cm^−1^ corresponded to the lattice vibration of the CaCO_3_ [33]. The ν4 in-plane bending mode of CaCO_3_ was observed at 712 cm^−1^ in some pores [34]. The shift at 1595 cm^−1^ was also attributed to CaCO_3_ [35].

By contrast, with 65% RH (Figure 4), the 828 and 894 cm^−1^ peaks were the characteristic of Raman shifts of C_3_S [36,37]. Apart from the signals of unhydrated C_3_S and the sheet, the 1084 cm^−1^ peak was prominent which was consistent with the results of Figure 3. The broad band at 551 cm^−1^ was due to Si-O-Si stretching modes in Q^3^ silicate tetrahedra [32]. The peak at 519 cm^−1^ resulted from the O-Si-O bending mode in the C-S-H [27,36]. 

Figure 5 is the Raman spectra of the control. Except for the peaks of unhydrated C_3_S (829 and 890 cm^−1^), the broad shoulder signal near 513 and 558 cm^−1^ were speculated to belong to the antisymmetric bending vibration of O-Si-O in unhydrated C_3_S [37,38]. The peak at 668 cm^−1^ was characteristic of Si-O-Si symmetrical bending vibration in Q^2^ silicate chains of C-S-H. The broad and strong peak at 1114 cm^−1^ was assigned to the hydroxylated species (Si-OH), which was due to the decrease in the proton numbers which bridged on Si-O of Q^2^ silicate tetrahedra [32,38]. The shift at 355 cm^−1^ was the characteristic band of Ca(OH)_2_ [37,38]. The signals around 1504 and 2234 cm^−1^ also belonged to Ca(OH)_2_ [39].

### 3.2. FT-IR

Figure 6 presents the FT-IR spectra of hydration products in the pores of the sheets. There were some characteristic peaks of unhydrated C_3_S in the hydrates of the 65% RH condition and the control. For instance, the 873 cm^−1^ band with shoulder bands at 921 and 835 cm^−1^ were related to the asymmetric stretching vibrations of Si-O bonds in unhydrated C_3_S [40,41]. Besides some peaks at 821, 873 and 920 cm^−1^, the absorption at 522 cm^−1^ was also attributed to unhydrated C_3_S [40]. In addition, the proximity signal at 804 cm^−1^ in the curve of water condition was due to Si-O-Si symmetric stretching of SiO_2_ [42,43].

The hydration results of water condition were similar with those of 65% RH condition. The absorption bands centered at 624 and 628 cm^−1^ of the two sheets were attributed to Si-O-Si bending vibration in depolymerized structure [44,45]. The broad absorption bands at 1107 cm^−1^ and 1103 cm^−1^ were both Si-O stretching vibrations in Q^3^ silicate tetrahedra [40,41,46]. However, the 968 cm^−1^ band observed in the control was the characteristic of Si-O stretching vibrations in Q^2^ units of C-S-H gel [40,46].

The peaks related to Ca(OH)_2_ and its carbonized products were found in all curves. The absorption at 871 cm^−1^ was the characteristic vibration mode of carbonate [41,46]. The bands at 1411, 1417 and 1483 cm^−1^ observed were characteristic of O-C=O symmetric stretching vibrations [47]. The peak at 1730 cm^−1^ was assigned to the C=O stretching vibrations which were related to the carbonation of Ca(OH)_2_ [45,48] In addition, The O-H bending vibration signals of water molecules were observed at 1643 cm^−1^ [46,49].

### 3.3. SEM/EDS

Figure 7 exhibits the SEM images of hydration products under different conditions. It can be seen that the hydration conditions played a prominent role on the morphology of the hydrates. The hydration products in the pore which was treated by the water condition were flake-like, while those treated by the 65% RH condition were honeycomb-like. The morphology of hydration products of the control exhibited interlaced fibers. 

The elements in the representative areas in Figure 7 were scanned by EDS and shown in Table 1. Besides the Ti contained in the sheet, the composition of different samples was similar, with calcium and oxygen in the majority. The calcium to silicon ratio (Ca/Si) of the hydration products hydrated by water condition was the highest, followed by that under normal conditions and treated by 65% RH condition. 

## 4. Discussion

The Raman shifts, FT-IR bands and their distributions observed in the hydration products are summarized in Table 2 and Table 3, respectively.

Combining the results of Raman and FT-IR spectra, it was clear that the hydration products exhibited a high degree of polymerization dominated by Q^3^ silicates for the hydration products in pores. By comparing the hydrates under the water condition and 65% RH condition, the amount of water filling the micropore played no obvious role on the polymeric structure. The difference was that there was still unhydrated C_3_S in the sample which was placed at the 65% RH condition.

SEM and EDS showed the morphology and their Ca/Si ratio of the hydration products in pores. A lower Ca/Si ratio was observed in the sample of the 65% RH condition than in that of the control. This is consistent with previous studies [50,51], which propose that the higher the polymerization degree, the lower the Ca/Si ratio. However, the hydrates under the water condition showed an unexpectedly higher Ca/Si ratio above 2. It lay in the fact that the C-S-H in this region was surrounded by Ca(OH)_2_ [52]. It can be confirmed that the oxygen content of products with water condition was higher than those at the 65% RH condition in those testing areas. The flake-like hydration products in this region were similar to Ca(OH)_2_ [53]. Although the results about the Ca/Si ratio of the products were inconsistent, it is certain that C-S-H gel formed in these micropores. The combinations of the Raman spectra, FT-IR spectra and SEM-EDS results demonstrated that the C-S-H grown in micropores had a high degree of polymerization.

The results of Raman and FT-IR spectra indicated that the C-S-H structure in control obviously grew as Q^2^ silicate tetrahedra. This finding has been confirmed by many scholars [38,40,54,55]. However, the hydration products in the pores showed a higher degree of polymerization (Q^3^), and there was almost no characteristic peak of Q^2^ unit. This finding indicates that space restriction affects the structure of the hydration products and the cement which hydrated in the space restriction of micropores tended to form C-S-H with silica tetrahedra of high polymerization degree.

Recently, the space filling hypothesis was opposed by Zajac et al. [56]. They proposed that the slow hydration in the later stage was not due to the effect of restricted space on the further growth of C-S-H, but caused by the slow transport of dissolved ions through the hydrates layer from the dissolution point to the precipitation point [56], and the slow diffusion had an impact on the evolution of the microstructures, such as the fine structure of the inner C-S-H [56]. In this paper, the characterization of the surface hydrates by means of Raman, FT-IR spectra and SEM-EDS indicated that space restriction actually changed the structure of the external hydrates. The influence of space restriction on the growth of C-S-H cannot be ignored, but the relationship between the structure change and the slow diffusion of dissolved ions needs to be further explored. At present, no theory can exclusively dominate the later hydration mechanism. The later stage kinetics is considered to be a combination of multiple mechanisms, such as diffusion mechanism, C-S-H densification and space filling hypothesis [2,13], which requires more research in the future.

## 5. Conclusions

This paper studied the microstructure of C_3_S hydrates in the restricted space of micropores in a sheet, by means of Raman, FT-IR spectra and SEM-EDS. Results indicate that space restriction affected the structure of the hydration products, and that the microstructure of the C-S-H which formed in the micropores was mainly composed of Q^3^ silicate tetrahedra with a high degree of polymerization. The amount of water filled in the micropore only affected the hydration degree, rather than the polymeric structure. The C-S-H which formed under conventional conditions mostly existed as a Q^2^ unit. The space restriction of cement hydration is conducive to the formation of C-S-H with silicate tetrahedra of a high polymerization degree.

## Figures and Tables

**Figure 1 materials-14-03645-f001:**
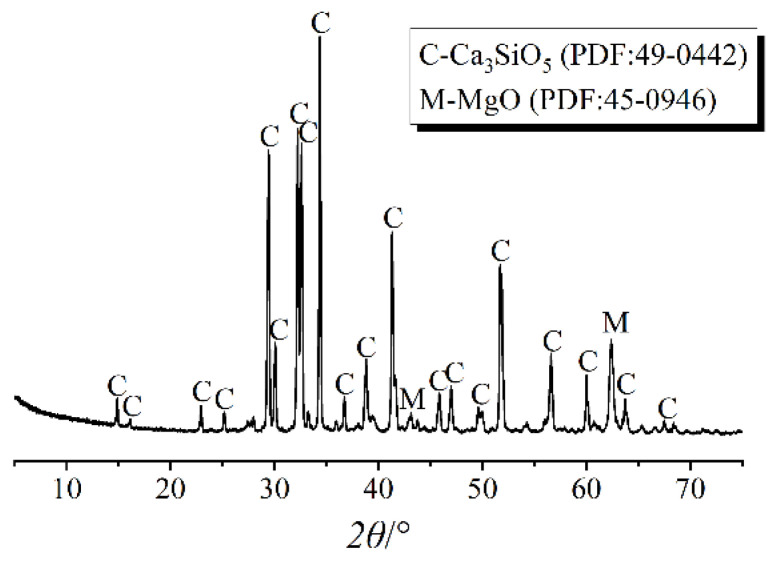
XRD patterns of C_3_S.

**Figure 2 materials-14-03645-f002:**
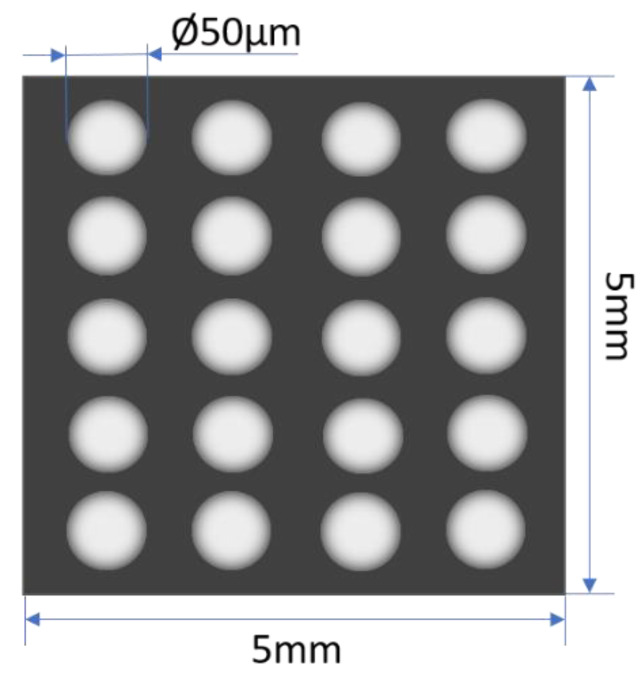
Schematic diagram of stainless steel sheet with pore array.

**Figure 3 materials-14-03645-f003:**
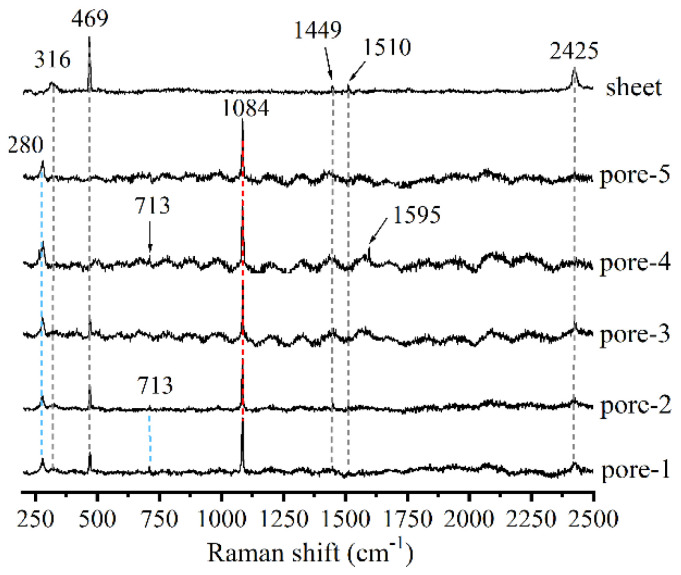
Raman spectra of hydration products in different pores (treated by water condition).

**Figure 4 materials-14-03645-f004:**
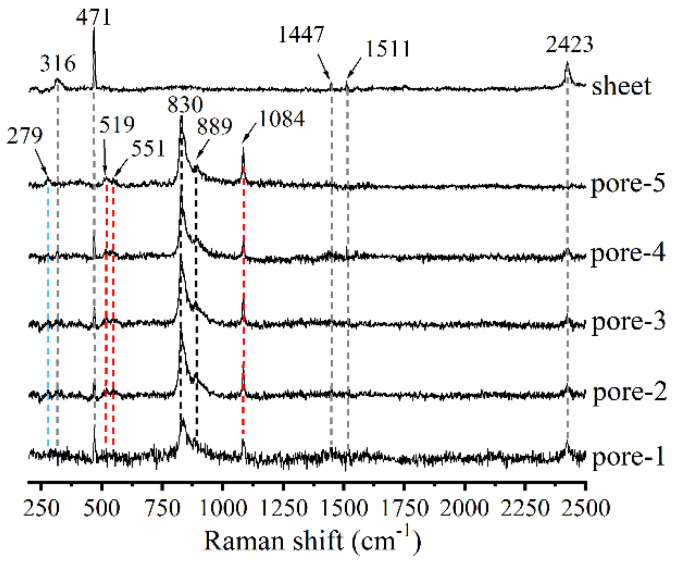
Raman spectra of hydration products in different pores (treated by 65% RH condition).

**Figure 5 materials-14-03645-f005:**
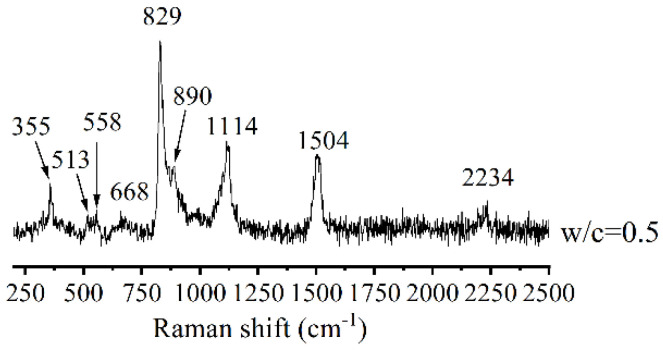
Raman spectra of hydration products (the control).

**Figure 6 materials-14-03645-f006:**
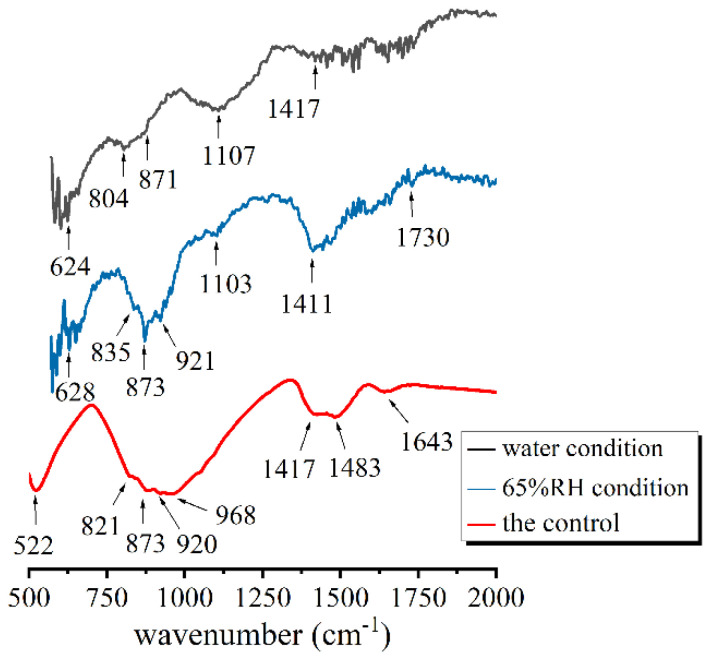
FT-IR spectra of hydration products under different hydration conditions.

**Figure 7 materials-14-03645-f007:**
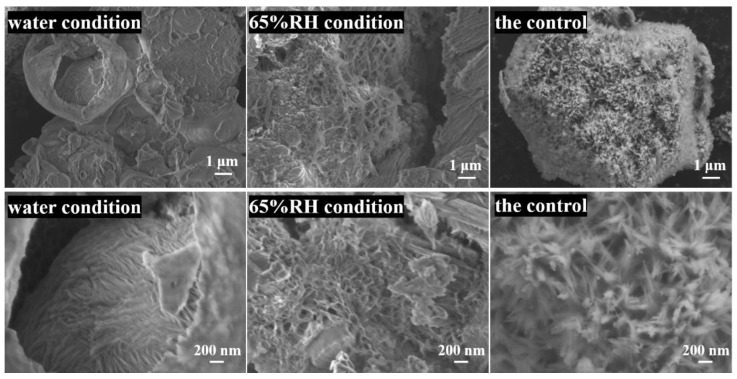
SEM images of hydration products under different conditions after 24 h.

**Table 1 materials-14-03645-t001:** EDX analysis of hydration products under different conditions after 24 h (at.%).

Elements	Ca	Si	O	C	Mg	Al	Ti	Ca/Si
water	22.8	8.6	60.9	5.9	0.9	0.3	0.6	2.65
65% RH	33.6	21.2	25.3	8.1	1.1	0.3	9.0	1.58
w/c = 0.5	20.1	10.6	58.8	9.7	0.7	0	0	1.89

**Table 2 materials-14-03645-t002:** Raman shifts and distribution of hydration products.

Conditions	Raman Shift (cm^−1^)				
	C_3_S	C-S-H	CaCO_3_	Ca(OH)_2_	Stainless Steel Sheet
water condition (pores)	——	1084 (Q^3^)	280, 713, 1595, 1581 and 1084	——	316, 469, 1449, 1510 and 2425
65% RH condition (pores)	830 and 889	519; 551 and 1084 (Q^3^)	279 and 1084	——	316, 471, 1417, 1511 and 2423
the control (w/c = 0.5)	513, 558, 829 and 890	668 (Q^2^); and 1114	——	355, 1504 and 2234	——

**Table 3 materials-14-03645-t003:** FT-IR bands and distribution of hydration products.

Conditions	FT-IR Spectra (cm^−1^)			
	C_3_S	C-S-H	CaCO_3_	Water
water condition (pores)	804	624; 1107 (Q^3^)	871 and 1417	——
65% RH condition (pores)	873, 921 and 835	628; 1103 (Q^3^)	1411 and 1730	——
the control (w/c = 0.5)	522, 821, 873 and 920	968 (Q^2^)	1417 and 1483	1643

## Data Availability

The data presented in this study are available on request from the corresponding author.

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
