# Peer review of "Impacts of Space Restriction on the Microstructure of Calcium Silicate Hydrate"

_materials, 2021, doi:10.3390/ma14133645_

Round 1

Reviewer 1 Report

The experimental work in the paper is good but it is no presented to show its importance.

English language is very poor, the paper needs and extensive review of language and grammar.

More details about the sample preparation is needed.

i cannot find any new contribution in the paper.

The discussion and analysis of the experimental work is poor.

The quality of the paper is very poor. The authors posed addressed the
topic of the paper as if it is a novel topic, while I didn't get any new
knowledge reading the paper. The paper lacks novelty. In answer to the
questions:
The paper is discussing the effect of space restriction on the formation
of C-S-H. The topic is interesting but the way it was tested and
presented does not add any value. It might be considered as a good topic
but this research did not address it properly. The research does not
have anything new as compared to the published research and the paper is
poorly written. Even the testing done is not explained properly when it
comes to its reasoning and conclusions.

Reviewer 2 Report

This is an interesting study of the calcium silicate hydrate growth under specific humidity and space restrictions. I have some remarks that should be addressed previous to the editor’s decision for acceptance.

  1. Lines 70-71. Could the authors provide more information regarding the preparation procedure of their material? Only the initial materials used are referred.
  2. Figure 1. Although I find no problem with it, maybe the authors should move it to the results section. Please, ignore this comment in the case that the other reviewers are ok with this.
  3. Lines 109-110. Could the authors interpret the sheet’s Raman peaks, with proper references?

  4. Lines 136-154 and Figure 6. I do have concerns regarding the interpretation of the FTIR measurements. Although the control spectrum is legit, the other two spectra present considerable noise. This is understandable and can be attributed to the humidity of the samples, but some of the identified peaks cannot be distinguished from noise, to my point of view. The problem lies with the 1701, 1539, 1417 and 624 cm-1 bands of the water condition and the 1581 and 628 cm-1 bands of the 65% RH condition.
  5. Table 3. Following the previous comment, CaCO3 does present bands at 1701 cm-1, but around 1790 cm-1
  6. Although FTIR and SEM-EDS are well-known techniques, the authors maybe should avoid abbreviations in the abstract.

Reviewer 3 Report

As it was compactly stated in the title, the paper presents the research on the impact of space restriction on the microstructure of calcium silicate hydrate. There were two sets of samples which were curried at temperature 20°C and 65% relative humidity. In the first set of samples, the sheet was covered by a drop of deionized water (water conditions) and the other one without addition of water (65% RH condition). I would like to ask the Authors to explain why they have decided for such currying conditions? The temperature is a standard value while testing Portland cement-based materials, however, the relative humidity usually is much higher (for example for strength test the required humidity while currying the samples is 95%). The whole research described in the paper would be of much greater practical importance, if it was adapted to the actual test conditions of Portland cement-based materials (OPC concrete, mortars etc.).

Regarding the figures in the paper (except Figure 2 and Figure 7) – they are probably presented as they were plotted from the devices, however, the descriptions are too small. They should be re-edited with the higher font.

Generally, the paper is written with the use of appropriate scientific vocabulary and good English. However, there are some minor errors – e.g. there is no such a country “America” (line 99) – probably it should be the USA or the United States of America.

And the final (general) remark. The paper reminds rather well-presented research report than the scientific paper. Moreover, the originality and novelty of research activity is not well exposed and/or highlighted. Presented conclusions are related to the performed laboratory tests - which is good of course. However, in the scientific paper there are also expected some general conclusions, indicating the contribution to the development of the relevant field of science.

Round 2

Reviewer 1 Report

I would like to thanks the authors for working on the paper. The paper was enhanced from the last version and is acceptable in the current form after reviewing the language.

Author Response

Point 1: I would like to thanks the authors for working on the paper. The paper was enhanced from the last version and is acceptable in the current form after reviewing the language.

Response 1: Thank you for your kind comments. We appreciate and acknowledge that your previous comments are valuable in improving the quality of our manuscript.

Reviewer 2 Report

Thank you for your feedback!

Author Response

Point 1: Thank you for your feedback!

Response 1: Thank you very much for your previous suggestion, which are valuable in improving the quality of our manuscript.